# MULAN: Multi-modal Causal Structure Learning and Root Cause Analysis for Microservice Systems

## ABSTRACT

Effective root cause analysis (RCA) is vital for swiftly restoring services, minimizing losses, and ensuring the smooth operation and management of complex systems. Previous data-driven RCA methods, particularly those employing causal discovery techniques, have primarily focused on constructing dependency or causal graphs for backtracking the root causes. However, these methods often fall short as they rely solely on data from a single modality, thereby resulting in suboptimal solutions.

In this work, we propose MULAN, a unified multi-modal causal structure learning method designed to identify root causes in microservice systems. We leverage a log-tailored language model to facilitate log representation learning, converting log sequences into time-series data. To explore intricate relationships across different modalities, we propose a contrastive learning-based approach to extract modality-invariant and modality-specific representations within a shared latent space. Additionally, we introduce a novel key performance indicator-aware attention mechanism for assessing modality reliability and co-learning a final causal graph. Finally, we employ random walks with restarts to simulate system fault propagation and identify potential root causes. Extensive experiments on three real-world datasets validate the effectiveness of our proposed framework.

## 1 INTRODUCTION

Root Cause Analysis (RCA) plays a critical role in identifying the origins of system failures, especially in microservice systems. A fault within any microservice can severely impact user experience and lead to substantial financial losses. To ensure the reliability and robustness of microservice systems, key performance indicators (KPIs) like latency, metrics data such as CPU/memory usage, and log data including pod-level Kubernetes entries are often collected and analyzed. However, the complexity of these systems combined with the vast amount of monitoring data can make manual root cause analysis both costly and error-prone. Thus, a swift and effective root cause analysis, enabling rapid service recovery and minimizing losses, is vital for the consistent operation and management of expansive, intricate systems.

Previous data-driven RCA studies, particularly those employing causal discovery techniques, have primarily focused on the construction of dependency/causal graphs. These graphs capture the causal relationships between various entities within a system and the associated KPIs so that the operators can trace back to the underlying causes by utilizing these established causal graphs. For instance, in [16], historical multivariate metric data was leveraged to construct causal graphs through conditional interdependence tests, followed by the application of causal intervention techniques to pinpoint the root causes within a microservice system. Furthermore, Wang et al. [39] introduced a hierarchical graph neural networks based approach to construct interdependent causal networks, facilitating the localization of root causes.

**Table 1: Abnormal patterns in multi-modal data for different system failures. '-' indicates no detected unusual patterns.**

| System Fault Type | System Metric | System Log |
| --- | --- | --- |
| Database Query Failure | - | Error/Warning |
| Login Failure | - | Error/Warning |
| DDoS Attack | High CPU Utilization | - |
| Disk Space Full | High Disk Utilization | Error/Warning |

However, these methods rely solely on data from a single modality, thus failing to capture the intricacies of various abnormal patterns associated with system failures, ultimately resulting in suboptimal solutions. Table 1 illustrates an example of anomalous information found in multi-modal data related to different types of system failures. Some system failures, such as Database Query Failures or Login Failures, may easily elude detection if we do not harness system logs to pinpoint their root causes. Conversely, system metrics and logs collectively contribute to the localization of system faults like "Disk Space Full". Leveraging multi-modal data empowers us to gain a deeper and more thorough insight into system failures, emphasizing the critical importance of adopting a more holistic approach to root cause analysis.

In recent years, multi-modal learning has emerged as a promising way in modeling diverse modalities across various domains, such as natural language processing [11, 20], information retrieval [10, 25], and computer vision [22, 30, 48]. Despite its prevalence, multi-modal learning for RCA is still largely unexplored. Recent multi-modal RCA approaches [13, 46] primarily aim to extract information from individual modalities, often missing the potential interplay between them. This oversight is particularly significant, given that studies on non-RCA multi-modal algorithms [44, 45] emphasize the pivotal role of harnessing relationships between modalities to optimize generalization outcomes.

Enlightened by multi-modal learning, this paper aims to propose a multi-modal causal structure learning method for identifying root causes in microservice systems. Formally, given the system KPI data and the multi-modal microservice data including metrics and log data, our goal is to learn a multi-modal causal graph to identify the top $k$ system entities that are most relevant to system KPI. There are three major challenges in this task:

- **C1: Learning effective representation of system logs for causal graph learning.** Traditional methods for learning causal graphs encounter difficulties when directly applied to system log data. Simply extracting statistical features overlooks the rich semantic information within the log messages. An intriguing approach is to employ language models to derive semantic insights. However, unstructured system logs significantly differ from standard textual data. They lack formal grammar rules and extensively employ special tokens. This divergence poses a considerable challenge when attempting to extract contextual information from log data using existing language models.

- **C2: Learning causal structure from multi-modal data.** Relying solely on the extraction of common information may inadvertently overlook critical insights unique to a single modality, potentially resulting in the failure to identify certain root causes. To enhance the applicability and robustness of the multi-modal RCA approach, the challenge is how to capture both modality-invariant information and modality-specific information and determine the corresponding effects associated with the system failure.
- **C3: Assessing modality reliability.** Data collected for root cause analysis often includes noisy metrics or overwhelming redundant log messages. This can obscure crucial patterns, making it a challenging task to identify significant events within the noise. However, existing methods typically treat both modalities equally important, thus suffering from the low-quality modality scenario. Consequently, it becomes imperative to re-weight the importance of each modality in noisy scenarios.

To tackle these challenges, in this paper, we propose MULAN, a MULti-Modal CAusal Structure LearNing method for root cause localization. MULAN consists of four major modules: 1) Representation Extraction via Log-Tailored Language Model; 2) Contrastive Multi-modal Causal Structure Learning; 3) Causal Graph Fusion with KPI-Aware Attention; and 4) Network Propagation based Root Cause Localization. Specifically, the initial step of MULAN is dedicated to extracting effective log representations, converting log sequences into time-series data to facilitate causal graph generation from system logs. To explore the relationships among different modalities, we introduce a contractive learning-based method that extracts both modality-invariant and modality-specific representations through node-level contrastive regularization and edge-level regularization. In the third module, a novel KPI-aware attention mechanism is designed to evaluate the reliability of each modality and fuse the final causal graph, ensuring the robustness of the root cause analysis model, especially in the presence of low-quality modalities. Finally, we employ random walk with restarts to simulate the propagation of a system fault and identify the root causes. Extensive experimental results with real-world datasets demonstrate the effectiveness of our proposed framework.

## 2 PRELIMINARIES

**Key Performance Indicator (KPI).** A KPI represents time series data that evaluates the efficiency and efficacy of a microservice architecture. For instance, latency and service response time are two common KPIs used in microservice systems. A large value of latency or response time usually indicates a low-quality system performance or even a system failure.

**Entity Metrics.** Entity metrics typically refer to the set of measurable attributes that give insight into the behavior and health of individual services (or entities) in a system. The system entity could be a physical machine, container, virtual machine, pod, *etc.* Some common entity metrics in a microservice system include CPU utilization, Memory utilization, disk IO utilization, *etc.* These entity metrics are essentially time series data. An abnormal system entity is usually a potential root cause of a system failure.

**Causal Structure Learning for Time Series Data.** Existing causal structure learning methods for time series data can be classified into four categories [2] including constrained-based methods [15, 26, 37], score-based methods [16, 21, 39], noise-based method [6, 19, 28], and other uncategorized methods [7, 14]. Our work belongs to the score-based category, which leverages the Vector Autoregression (VAR) Model [34] to model multi-modal causal relationships between different system entities.

One branch of score-based methods [16, 27, 39] aim to utilize $p$-th order VAR Model to capture the relationship between different system entities as they change over time. Given the $T$-length time-series data $X = \{x_0, ..., x_T\}$, these methods utilize the $p$-th order data before the $t$-th timestamp to predict the value at the timestamp $t$ via the VAR model as follows:

$$x_t = A_1 x_{t-1} + \cdots + A_p x_{t-p} + \epsilon_t$$

where $x_t \in \mathbb{R}^{n-1}$, $n-1$ is the number of entities, $p$ denotes the time-lagged order, $A_p \in \mathbb{R}^{n-1 \times n-1}$ is the weight matrix at the $p$-th time-lagged order, and $\epsilon_t \in \mathbb{R}^{n-1}$ is the error variables. The underlying intuition is to predict the future value at the $t$-th timestamp by utilizing the last $p$-length historical values.

Inspired by the message-passing mechanism of the graph neural network [18, 38], [39] combined the VAR model with the graph neural network to capture the non-linear relationship between different system entities by:

$$\tilde{X} = f(\sum_p A \hat{X}_p; \theta) + \epsilon \quad (1)$$

where $\tilde{X} \in \mathbb{R}^{(n-1) \times m}$ denotes the future data, $m = T - p + 1$ denotes the length of the effective timestamp, $A \in \mathbb{R}^{(n-1) \times (n-1)}$ is the learnable weight matrix shared across different $p$, $\hat{X}_p \in \mathbb{R}^{(n-1) \times m}$ is the $p$-lagged historical data and $\theta$ is the parameters of the graph neural networks $f$. Notice that different from the traditional graph neural network where the adjacency matrix is given, in Equation. 1, $A$ is also a learnable adjacency matrix aiming to capture the non-linear relationship between system entities. Therefore, [39] aimed to minimize the following loss:

$$\min(||\tilde{X} - f(\sum_p A \hat{X}_p; \theta)||^2) \quad (2)$$

Note that these methods [27, 39] are designed for one single modality and simply extending it to include multiple modalities would result in the sub-optimal performance, which is validated in the experiment (*i.e.*, Subsection 4.2.1 and Subsection 4.2.2).

**Problem Statement.** Let $\mathcal{X}^M = \{X_0^M, ..., X_a^M\}$ denotes a multivariate time series metric data. The $i$-th metric data is $X_i^M = [x_{i,0}^M, ..., x_{i,T}^M] \in \mathbb{R}^{(n-1) \times T}$, the unstructured system logs $X^L$, and system key performance indicator $\mathbf{y} \in \mathbb{R}^T$, the goal is to construct a causal graph $\mathcal{G} = \{V, A\}^*$ to identify the top $k$ system entities that are most relevant to $\mathbf{y}$, where $V$ is the set of vertices, $A \in \mathbb{R}^{n \times n}$ is the adjacency matrix, $n$ is the number of entities plus the system KPI, and $T$ is the length of time series. For simplicity, we concatenate the $i$-th system metric and KPI together $X_i^M = [X_i^M; \mathbf{y}] \in \mathbb{R}^{n \times T}$ [†] to illustrate our model.

---

[*]Note that the causal graph $\mathcal{G}$ consists of two types of nodes, including the system entities and the system KPI.

[†]For ease of explanation, we use one system metric to introduce our model.

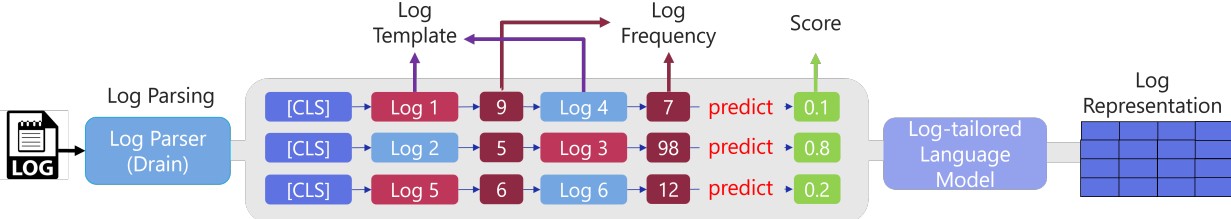

**Figure 1: The overview of the proposed framework MULAN. It consists of four main modules: representation extraction via log-tailored language model, contrastive multi-modal causal structure learning, causal graph fusion with KPI-aware attention, and network propagation-based root cause localization.**

**Figure 2: The overview of log representation extraction. It first uses a log parser to extract the log templates. The inputs of the language model are log sequences, where unique log templates are followed by their frequencies within a fixed time window. The label information (*i.e.*, scores) are obtained through anomaly detection methods to guide the log sequence representation learning. [CLS] is a special token used for downstream tasks.**

## 3 METHODOLOGY

We present MULAN, a multi-modality causal structural learning method for root cause analysis. As illustrated in Figure 1, MULAN includes four key modules: (1) representation extraction via log-tailored language model; (2) contrastive multi-modal causal structure learning; (3) causal graph fusion with KPI-aware attention; and (4) network propagation based root cause localization.

### 3.1 Representation Extraction via Log-tailored Language Model

The first step of MULAN is to transform raw system logs into time series data, making it easier to generate causal graphs from these logs. The main challenge is how to effectively learn high-quality representations from these unstructured system logs (*i.e.*, challenge **C1** in Section 1). A straightforward method involves fine-tuning a pre-trained large language model with system logs to generate representations for log sequences. However, it's essential to understand that system logs are quite different from traditional textual data. They lack formal grammar rules, make extensive use of special tokens, and lack a structured format, which makes it challenging to extract the necessary contextual information. As a result, merely fine-tuning pre-trained language models on system logs often leads to suboptimal representations. On the other hand, it's vital to extract semantic information from log event content to obtain high-quality representations[47]. Unfortunately, most existing approaches [8, 43] fail to capture such important information, thus suffering from the performance degradation.

To address the challenge **C1**, we introduce a log-tailored language model (as illustrated in Figure 2). This model unfolds in three key phases:

**Phase 1**: We utilize an existing log parsing tool (*e.g.*, Drain) to transform unstructured system logs into structured log messages, represented as log templates.

**Phase 2**: The entire system logs are partitioned into multiple time windows with fixed sizes. For each time window, we assemble a log sequence $X_{i,j}^L, j \in [0, T]$ for the $i$-th entity. These sequences consist of unique log templates that occur within that specific time range. Rather than treating individual words in log templates as tokens, we treat each event template as a token, and the log templates within a sequence are organized based on their first appearance timestamp in ascending order. This strategy significantly reduces the token count and minimizes the maximum sequence length, which speeds up the training process. Moreover, it enables the encoding of semantic information into representations and models the relationships among log templates within a log sequence. We also consider the frequency of each unique log template, assuming that more frequently occurring log event templates carry more important information. This assumption proves useful in dealing with certain failure cases, such as DDoS attacks. In the event of a DDoS attack, the frequency of certain log templates may suddenly and dramatically increase, indicating unusual behavior. To address this, we include the frequency right after each log template, providing extra information for monitoring unusual patterns in potential failure cases. Additionally, we leverage log-based anomaly detection

algorithms (*e.g.*, OC4Seq [43] or Deeplog [8]) to measure the anomaly score denoted as $y^{log}$. This score is used as label information to assist language models in learning better representations. The objective function is formulated as follows:

$$\mathcal{L}_{log} = \mathbb{E}_{i,j}||y_i^{log} - f(X_{i,j}^L, c_{i,j}^L)||^2 \qquad (3)$$

where $c_{i,j}^L$ denotes a list of the frequency of the unique log templates within a log sequence $X_{i,j}^L$, and $f(\cdot)$ is the proposed language model that predicts the anomaly score. The details of extracting high-quality label information can be found in Appendix A.

**Phase 3**: We train a regression-based language model by optimizing $\mathcal{L}_{log}$ (Eq. 3). The core of our proposed language model consists of a bidirectional transformer, followed by a one-layer multilayer perceptron for predicting the anomaly scores. We then employ this model to generate the representation of the special token [CLS], which serves as the representation of the log sequence $x_{i,j}^L \in \mathbb{R}^d$ for the $i$-th entity at the $j$-th time window. Note that the feature dimension, typically $d = 768$ in traditional large language models, can be further reduced using feature reduction techniques like PCA [1]. This reduction can potentially bring the dimension down to a much lower value (*e.g.*, $d = 1$ in this paper), facilitating the causal structure learning process. Therefore, for the $i$-th system entity, we obtain $\hat{X}_i^L = \{x_{i,0}^L, ..., x_{i,T}^L\} \in \mathbb{R}^T$. The structured log representations for the $n - 1$ other system entities are denoted as $\hat{X}^L = [\hat{X}_0^L; ...; \hat{X}_{n-1}^L] \in \mathbb{R}^{(n-1)\times T}$. Similarly, we concatenate the system log and KPI together $\hat{X}^L = [\hat{X}^L; y] \in \mathbb{R}^{n\times T}$.

## 3.2 Contrastive Multi-modal Causal Structure Learning

As previously mentioned, existing methods [23, 29, 39, 46] often struggle to handle multi-modal data or fail to effectively correlate different modalities, resulting in suboptimal solutions. Furthermore, exclusively extracting modality-invariant information may lead to a loss of valuable insights within individual modalities. To address this challenge (*i.e.*, challenge **C2** in Section 1), we propose a contrastive learning based method for extracting both the modality-invariant representation and the modality-specific representation via encoder-decoder pairs.

**Contrastive Learning-based Encoders**. Given the system metric data $\hat{X}^M = X^M \in \mathbb{R}^{n\times T}$ and the system log representation $\hat{X}^L \in \mathbb{R}^{n\times T}$, we first extract both modality-invariant and modality-specific representation by:

$$
\begin{aligned}
R_c^v &= E_c^v(\hat{X}^v, A^v) \\
R_s^v &= E_s^v(\hat{X}^v, A^v) \\
R_c &= \alpha R_c^L + (1 - \alpha) R_c^M \\
H^v &= MLP^v(R_c^v)
\end{aligned}
\qquad (4)
$$

where $v \in \{M, L\}$ represents the two modalities, $R_c^v \in \mathbb{R}^{n\times m\times d_1}$ denotes the modality-invariant representation extracted from the $v$-th modality, $R_s^v \in \mathbb{R}^{n\times m\times d_1}$ represents the modality-specific representation from the $v$-th modality, and $m$ is the length of the effective timestamps. $R_c$ is the combined modality-invariant representation, $H^v \in \mathbb{R}^{n\times d_2}$ can be viewed as the entity representations, while $d_1$

and $d_2$ denote the hidden feature dimensions. And $\alpha \in [0, 1]$ is a constant parameter balancing two modality-invariant representations.

Here, we employ GraphSage [12] as the backbone for both $E_c^v$ and $E_s^v$ to extract modality-invariant representation and modality-specific representation, respectively. And $MLP^v(\cdot)$ is a Multi-Layer Perception (MLP) used to map the representation $R_c^L$ and $R_c^M$ to another latent space to get the entity representations. It's important to note that, in contrast to traditional graph neural networks, where the adjacency matrix is predefined, in Eq. 4, $A^v$ is also a learnable adjacency matrix designed to capture the non-linear relationships among system entities.

To ensure mutual information agreement between the modality-invariant representations extracted from both metric and log data, we propose maximizing the mutual information between these two representations using contrastive learning regularization:

$$\mathcal{L}_{node} = -\frac{1}{n}\sum_{i=1}^{n} \frac{sim(H_i^M, H_i^L)}{\sum_k sim(H_i^M, H_k^L)} \qquad (5)$$

where $sim(H_i^M, H_k^L) = \frac{H_i^M(H_k^L)^T}{|H_i^M||H_k^M|}$ is the cosine similarity measurement between two entity representations $H_i^L$ and $H_k^M$.

To ensure that there is no information overlapping between the modality-invariant and modality-specific representations, we leverage the orthogonal constraint [41], defined as:

$$\mathcal{L}_{orth} = \sum_{v\in\{M,L\}}\sum_{i=1}^{n} ||(R_{s,i}^v)^T R_{c,i}^v||_F^2 \qquad (6)$$

However, minimizing $\mathcal{L}_{node}$ and $\mathcal{L}_{orth}$ alone cannot guarantee that the modality-invariant representations contain only information relevant to learning causal graphs. To further ensure the quality of modality-invariant representations, we propose predicting the adjacency matrix of the causal graph based on the representation of edges as follows:

$$\mathcal{L}_{edge} = \sum_{v\in\{M,L\}}\sum_{i,j} ||G(e_{ij}^v) - A_{ij}^v||^2 \qquad (7)$$

where $e_{ij}^v = [H_i^v, H_j^v]$ denotes the concatenation of the representations of two entities, and $G(\cdot)$ is a one-layer MLP followed by the sigmoid activation function used to predict the existence of an edge in $A^v$. Note that the causal graph (*e.g.*, $\mathcal{G}^v = \{V, A^v\}$) includes both the system entities and the system KPI. Encoding the topological structure of the causal graph in Eq. 7 allows us to better capture the relationship between the root causes and the system KPI.

**VAR-based Decoders**. After extracting both modality-invariant and modality-specific representations, we aim to predict the future value $\tilde{X}^v$ with the previous $p$-th lagged data $\hat{X}^v$ via VAR model:

$$\mathcal{L}_{var} = \sum_{v\in\{M,L\}} ||\tilde{X}^v - D^v(R_c + R_s^v)||^2 \qquad (8)$$

Similarly, we choose GraphSage as the backbone for the decoder $D^v(\cdot)$.

## 3.3 Causal Graph Fusion with KPI-Aware Attention

From the metric decoder and log decoder, we can obtain the causal graph $\mathcal{G}^M$ and the causal graph $\mathcal{G}^L$, respectively. Combining these two causal graphs through simple addition is not suitable, as it may lead to dense and cyclical graphs. Furthermore, in scenarios with low-quality modalities (as discussed in challenge **C3** in Section 1), treating both modalities as equally important would yield undesirable results. Following the assumption that KPI is highly associated with the root cause [39], we propose a KPI-aware attention-based causal graph fusion. This module measures the cross-correlation [3] between the raw feature of each entity for each modality and the KPI to alleviate the potential negative impact of low-quality modalities by:

$$s^v = \max_{p \in [0,\tau]} (X^v \odot \mathbf{y})(p) = \max_{p \in [0,\tau]} \int_{t=0}^{+\infty} X^v(t+p) \cdot \mathbf{y}(t) dt \quad (9)$$

where $p$ represents the time lag and $\tau$ is the maximum time lag. Intuitively, $s^v$ quantifies the maximum similarity between each entity and the KPI while considering a time lag of up to $\tau$. A higher value of $s^v$ indicates a stronger causal relationship between the system entity and the KPI.

Given that the temporal patterns of the top $k$ entities within a high-quality modality are expected to closely resemble the temporal pattern of the KPI, and that smaller values of $s^v$ typically indicate low-quality modalities, we employ $s^v$ to measure the importance of each modality as follows:

$$a^v = \sigma\left(\sum_{i \in idx^v} s_i^v\right) = \frac{e^{\sum_{i \in idx^v} s_i^v}}{e^{\sum_{i \in idx^L} s_i^L} + e^{\sum_{i \in idx^M} s_i^M}} \quad (10)$$

$$idx^v = \text{Topk}(s^v[-1,:];k)$$

where $\sigma(\cdot)$ is the softmax function. We validate this assumption in Section 4.2.2. Notably, we can leverage the modality importance score $a^v$ to replace the hyper-parameter $\alpha$ in Eq. 4 and get the final fused adjacency matrix for the causal graph $\mathcal{G}$ as follows:

$$R_c = a^L R_c^L + a^M R_c^M$$
$$A = a^L A^L + a^M A^M \quad (11)$$

**Optimization.** Therefore, the final objective function is written as:

$$\mathcal{L} = \lambda_1 \mathcal{L}_{var} + \lambda_2 \mathcal{L}_{orth} + \lambda_3 \mathcal{L}_{node} + \lambda_4 \mathcal{L}_{edge} + \lambda_5 ||A||_1 + h(A) \quad (12)$$

where $||\cdot||_1$ is the sparsity constraint imposed on the adjacency matrix. The trace exponential function $h(A) = (tr(e^{A*A}) - n) = 0$ holds if and only if $A$ is acyclic [27], where $*$ denotes Hadamard product of two matrices. $\lambda_1$, $\lambda_2$, $\lambda_3$, $\lambda_4$, and $\lambda_5$ are the positive constant hyper-parameters. The parameter analysis can be found in Subsection 4.2.3.

## 3.4 Network Propagation based Root Cause Localization

The final fused causal graph $\mathcal{G} = \{V, A\}, A \in \mathbb{R}^{n \times n}$ consists of two types of nodes: system entities and system KPI. Malfunctioning effects can propagate from the root cause to its neighboring entities, meaning that the first-order neighbors of system KPIs may not necessarily be the root causes. To pinpoint the root cause, we first derive the transition probability matrix based on the causal graph $\mathcal{G}$ and then employ a random walk with restart method [36] to mimic the propagation patterns of malfunctions. Specifically, the transition probability matrix $P$ is formulated as follows:

$$P_{ij} = \frac{(1-\beta) A_{j,i}}{\sum_{k=1}^{n} A_{k,i}} \quad (13)$$

where $\beta \in [0,1]$ represents the probability of transitioning from one node to another. The probability transition equation for the random walk with restart is given by:

$$\mathbf{p}_{t+1} = (1-c)\mathbf{p}_t + c\mathbf{p}_0 \quad (14)$$

where $\mathbf{p}_t$ denotes the jumping probability at the $t$-th step, $\mathbf{p}_0$ is the initial starting probability, and $c \in [0,1]$ is the restart probability. Once the jumping probability $\mathbf{p}_t$ converges, the probability scores of the nodes are used to rank the system entities. The top $k$ entities are then selected as the most likely root causes for the system failure.

## 4 EXPERIMENTS

In this section, we evaluate the effectiveness of our proposed MULAN through a comparative analysis with state-of-the-art root cause analysis methods. Additionally, we conduct a comprehensive case study and an ablation study to further validate the assumptions outlined in the Methodology section.

### 4.1 Experiment Setup

*4.1.1 Datasets.* We evaluate the performance of our method, MULAN, using three real-world datasets for root cause analysis: (1). **Product Review** [39]: This microservice system, dedicated to online product reviews, encompasses 234 pods and is deployed across 6 cloud servers. It recorded four system faults between May 2021 and December 2021. (2). **Online Boutique** [46]: This dataset represents a microservice system designed for e-commerce, and it includes five system faults. (3). **Train Ticket** [46]: This dataset is a microservice system for railway ticketing service with 5 system faults. All three datasets contain two modalities: system metrics and system logs.

*4.1.2 Evaluation Metrics.* To measure the model performance, we choose three widely-used metrics [24, 39]: (1). **Precision@K (PR@K)**: This metric measures the probability that the top-K predicted root causes are accurate. (2). **Mean Average Precision@K (MAP@K)**: It provides an assessment of the top-K predicted causes from an overall perspective. (3). **Mean Reciprocal Rank (MRR)**: This metric evaluates the ranking capability of the models. The details of these three metrics can be found in Appendix B.2.

*4.1.3 Baselines.* We compare MULAN with six causal discovery models: (1). **PC** [4]: This classic constraint-based causal discovery algorithm is designed to identify the causal graph's skeleton using an independence test. (2) **Dynotears** [27]: It constructs dynamic Bayesian networks through vector autoregression models. (3). **C-LSTM** [35]: This model utilizes LSTM to model temporal dependencies and capture nonlinear Granger causality. (4). **GOLEM** [26]: GOLEM relaxes the hard Directed Acyclic Graph (DAG) constraint of NOTEARS [49] with a scoring function. (5). **REASON** [39]: An

**Table 2: Results on Product Review dataset w.r.t different metrics.**

| Modality | Model | PR@1 | PR@5 | PR@10 | MRR | MAP@3 | MAP@5 | MAP@10 |
|---|---|---|---|---|---|---|---|---|
| Metric Only | Dynotears | 0 | 0 | 0.50 | 0.070 | 0 | 0 | 0.075 |
| | PC | 0 | 0 | 0.25 | 0.053 | 0 | 0 | 0.050 |
| | C-LSTM | 0.25 | 0.75 | 0.75 | 0.474 | 0.5 | 0.25 | 0.675 |
| | GOLEM | 0 | 0 | 0.25 | 0.043 | 0 | 0 | 0.025 |
| | REASON | 0.75 | **1.0** | **1.0** | 0.875 | 0.917 | 0.95 | 0.975 |
| Log Only | Dynotears | 0 | 0 | 0.25 | 0.058 | 0 | 0 | 0.075 |
| | PC | 0 | 0 | 0.25 | 0.069 | 0 | 0 | 0.075 |
| | C-LSTM | 0 | 0 | 0.25 | 0.059 | 0 | 0 | 0.075 |
| | GOLEM | 0 | 0 | 0.25 | 0.058 | 0 | 0 | 0.075 |
| | REASON | 0 | 0.50 | 0.75 | 0.216 | 0.167 | 0.25 | 0.400 |
| Multi-Modality | Dynotears | 0 | 0 | 0.50 | 0.095 | 0 | 0 | 0.150 |
| | PC | 0 | 0 | 0.25 | 0.064 | 0 | 0 | 0.125 |
| | C-LSTM | 0.50 | 0.75 | 0.75 | 0.592 | 0.583 | 0.65 | 0.700 |
| | GOLEM | 0 | 0 | 0.25 | 0.065 | 0 | 0 | 0.050 |
| | REASON | 0.75 | **1.0** | **1.0** | 0.875 | 0.917 | 0.95 | 0.975 |
| | Nezha | 0 | 0.5 | 0.75 | 0.193 | 0.083 | 0.25 | 0.475 |
| | MULAN | **1.0** | **1.0** | **1.0** | **1.0** | **1.0** | **1.0** | **1.0** |

**Table 3: Results on Online Boutique dataset w.r.t different metrics.**

| Modality | Model | PR@1 | PR@3 | PR@5 | MRR | MAP@2 | MAP@3 | MAP@5 |
|---|---|---|---|---|---|---|---|---|
| Metric Only | Dynotears | 0.20 | 0.40 | 0.40 | 0.344 | 0.20 | 0.267 | 0.320 |
| | PC | 0.20 | 0.40 | 0.80 | 0.390 | 0.30 | 0.333 | 0.400 |
| | C-LSTM | 0 | 0.40 | 0.80 | 0.30 | 0.10 | 0.200 | 0.440 |
| | GOLEM | 0 | 0.40 | 0.80 | 0.291 | 0.20 | 0.267 | 0.360 |
| | REASON | 0.40 | 0.80 | 1.0 | 0.617 | 0.50 | 0.200 | 0.440 |
| Log Only | Dynotears | 0 | 0.20 | 0.60 | 0.207 | 0 | 0.067 | 0.240 |
| | PC | 0 | 0.40 | 0.60 | 0.257 | 0.10 | 0.200 | 0.320 |
| | C-LSTM | 0 | 0.40 | 0.60 | 0.267 | 0.10 | 0.200 | 0.360 |
| | GOLEM | 0 | 0.40 | 0.80 | 0.248 | 0 | 0.133 | 0.360 |
| | REASON | 0.20 | 0.80 | 0.80 | 0.458 | 0.30 | 0.467 | 0.600 |
| Multi-Modality | Dynotears | 0.20 | 0.60 | 1.0 | 0.467 | 0.30 | 0.400 | 0.640 |
| | PC | 0.40 | 0.80 | 1.0 | 0.573 | 0.40 | 0.533 | 0.680 |
| | C-LSTM | 0.20 | 0.40 | 1.0 | 0.450 | 0.30 | 0.333 | 0.600 |
| | GOLEM | 0.20 | 0.60 | 1.0 | 0.467 | 0.30 | 0.400 | 0.640 |
| | REASON | 0.40 | 1.0 | 1.0 | 0.667 | 0.60 | 0.733 | 0.840 |
| | Nezha | 0.60 | 1.0 | 1.0 | 0.767 | 0.70 | 0.800 | 0.880 |
| | MULAN | **0.80** | **1.0** | **1.0** | **0.900** | **0.90** | **0.933** | **0.960** |

interdependent network model that focuses on learning both intra-level and inter-level causal relationships. (6). **Nezha** [46]: A multi-modal method designed to identify root causes by detecting abnormal patterns.

## 4.2 Performance Evaluation

*4.2.1 Experimental Results.* Tables 2, 3, and 4 present a comprehensive performance evaluation of all methods. For methods exclusively designed for a single modality (*e.g.*, PC, C-LSTM, REASON, Dynotears, and GOLEM), we assess their performance in both single-modality scenarios (*e.g.*, system metrics only or system logs only) and the multi-modality case. To enable multi-modality modeling, we initially convert the system logs into time-series data using the Regression-based language model introduced in Section 3.1. This time-series data is then treated as an additional system metric for evaluation. We calculate an average ranking score based on the evaluation of different system metrics as the final result for all single-modality methods and MULAN. Our observations are as follows: (1) In contrast to single-modality scenarios, most baseline methods demonstrate improved performance when leveraging multi-modality data across three distinct datasets and various

metrics. (2) MULAN consistently outperforms all baseline methods across the three datasets. Notably, MULAN exhibits a remarkable improvement in MRR on the Product Review dataset, surpassing the second competitor (*i.e.*, REASON) by 12.5%. Furthermore, on the Online Boutique dataset, MULAN outperforms Nezha, achieving improvements of over 13.2% and 8% with respect to MRR and MAP@5, respectively. This superiority can be attributed to MULAN's adeptness in exploring correlations among different modalities and its robust KPI-aware attention mechanism, while all baseline methods, including REASON and Nezha, fall short in this regard.

*4.2.2 Case Study.* In this case study, we aim to demonstrate the robustness of our proposed method in the context of low-quality modality scenarios. It's important to distinguish this setup from the experimental configuration detailed in Section 4.2.1, where multiple system metrics were utilized. In this case study, we keep the representation of the system log constant and only select a single system metric to investigate its impact on the performance of all models. The procedure unfolds as follows: Initially, we assess the performance of each single-modality baseline method using distinct system metrics (*e.g.*, CPU usage, memory usage, transmit

**Table 4: Results on Train Ticket Dataset *w.r.t.* Different Metrics.**

| Modality | Model | PR@1 | PR@5 | PR@10 | MRR | MAP@3 | MAP@5 | MAP@10 |
|---|---|---|---|---|---|---|---|---|
| | Dynotears | 0 | 0 | 0.2 | 0.046 | 0 | 0 | 0 |
| | PC | 0 | 0.2 | 0.8 | 0.170 | 0.133 | 0.16 | 0.243 |
| Metric Only | C-LSTM | 0 | 0.2 | 0.4 | 0.096 | 0 | 0 | 0.100 |
| | GOLEM | 0 | 0.2 | 0.4 | 0.098 | 0 | 0 | 0.100 |
| | REASON | **0.2** | **0.6** | 0.8 | 0.323 | 0.2 | 0.28 | 0.343 |
| | Dynotears | 0 | 0.4 | 0.8 | 0.160 | 0 | 0.16 | 0.271 |
| | PC | 0 | 0.4 | 0.8 | 0.219 | 0.133 | 0.24 | 0.343 |
| Log Only | C-LSTM | 0 | 0.4 | 0.8 | 0.160 | 0 | 0.16 | 0.271 |
| | GOLEM | 0 | 0.4 | 0.8 | 0.164 | 0 | 0.16 | 0.274 |
| | REASON | **0.2** | 0.4 | **1.0** | 0.315 | 0.2 | 0.28 | 0.343 |
| | Dynotears | 0 | 0.4 | 0.8 | 0.141 | 0 | 0.16 | 0.228 |
| | PC | 0 | 0 | 0.4 | 0.083 | 0 | 0 | 0.071 |
| | C-LSTM | **0.2** | 0.4 | 0.8 | 0.310 | 0.2 | 0.28 | 0.314 |
| Multi-Modality | GOLEM | 0 | 0.4 | 0.8 | 0.160 | 0 | 0.16 | 0.271 |
| | REASON | **0.2** | 0.4 | 0.8 | 0.299 | 0.2 | 0.28 | 0.300 |
| | Nezha | **0.2** | 0.2 | **1.0** | 0.297 | 0.2 | 0.2 | 0.285 |
| | MULAN | **0.2** | 0.4 | **1.0** | **0.381** | **0.333** | **0.36** | **0.414** |

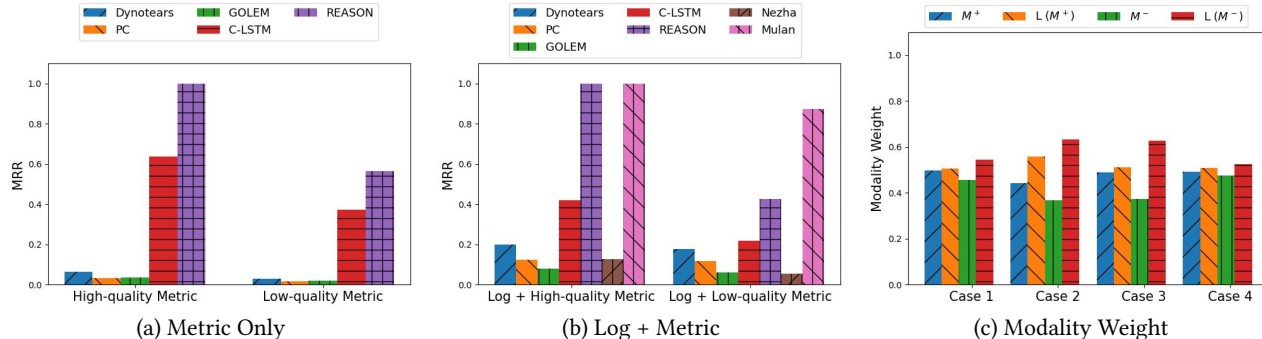

(a) Metric Only  (b) Log + Metric  (c) Modality Weight

**Figure 3: Case study on Product Review dataset. (a): MRR score of all methods evaluated with a single system metric only. (b): MRR score of all methods evaluated with one system metric and system log. (c): Modality weight measured by KPI-aware mechanism of MULAN with four system fault cases, where $M^+$, $L(M^+)$, $M^-$, and $L(M^-)$ are the weight of the high-quality metric, the weight of the system log with the high-quality metric, the weight of the low-quality metric and the weight of the system log with the low-quality metric, respectively.**

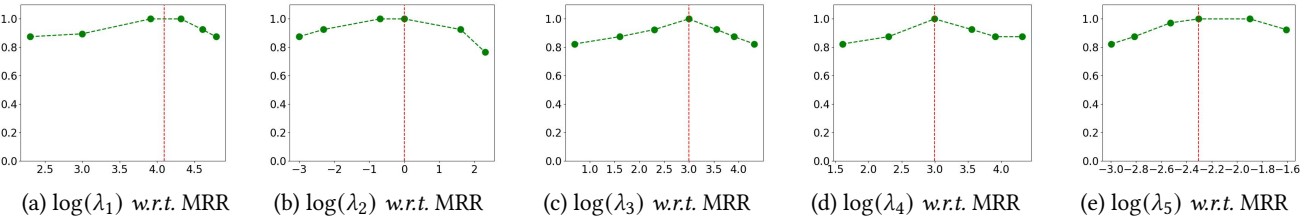

(a) $\log(\lambda_1)$ *w.r.t.* MRR  (b) $\log(\lambda_2)$ *w.r.t.* MRR  (c) $\log(\lambda_3)$ *w.r.t.* MRR  (d) $\log(\lambda_4)$ *w.r.t.* MRR  (e) $\log(\lambda_5)$ *w.r.t.* MRR

**Figure 4: Parameter analysis on Product Review dataset w.r.t MRR. The red dashed line denotes the value used in Table 2.**

rate, *etc*). Subsequently, we identify the system metric with the highest median ranking score as the high-quality metric, denoted as $M^+$, and the system metric with the lowest median ranking score as the low-quality metric, denoted as $M^-$. Detailed results of the ranking score are available in Appendix B.3. We evaluate all methods' performance on the Product Review Dataset and present the results in Figures 3 (a) and (b). Notably, the performance of the log-only setting can be found in Table 2. Additionally, to underscore the robustness of MULAN, we also examine the weights assigned to the two modalities, as illustrated in Figure 3 (c).

We have four key observations: (1). REASON and C-LSTM demonstrate impressive results on the Product Review dataset when utilizing a high-quality metric. However, their performance undergoes a significant decline when the high-quality metric is substituted with the low-quality system metric, as shown in Figure 3 (a). (2). Comparing Figures 3 (a) and 3 (b) reveals that the performance of most baseline methods improves when incorporating system logs, regardless of whether the chosen system metric is of high or low quality. This underscores the value of integrating multiple modalities for enhanced performance. (3). In Figure 3 (b), the performance of most baseline methods experiences a notable decrease

**Table 5: Ablation study on three datasets w.r.t MRR.**

| Model | Product Review | Online Boutique | Train Ticket |
|---|---|---|---|
| MULAN | **1.0** | **0.9** | **0.347** |
| MULAN-V | 0.875 | 0.8 | 0.343 |
| MULAN-O | 0.833 | **0.9** | 0.326 |
| MULAN-N | 0.813 | 0.7 | 0.343 |
| MULAN-E | 0.833 | 0.667 | 0.225 |

when replacing the high-quality metric with the low-quality one. Remarkably, our proposed method (MULAN) consistently maintains robust and promising performance in such scenarios. (4) In Figure 3 (c), when the high-quality system metric ($M^+$ or blue bar) is replaced by the low-quality system metric ($M^-$ or green bar), MULAN dynamically adjusts the weight assigned to the system metric in all four cases. This adaptability ensures that MULAN does not overly rely on any specific metric modality. Findings 3 and 4 underscore the effectiveness of the KPI-aware attention mechanism and the inherent robustness of the proposed method.

*4.2.3 Parameter Analysis.* In this subsection, we conduct a thorough analysis of the parameter sensitivity of MULAN framework on the Product Review dataset. For a detailed parameter sensitivity analysis of the Online Boutique and Train Ticket datasets, please refer to Appendix B.4. We specifically examine the impact of variations in five parameters: $\lambda_1$, $\lambda_2$, $\lambda_3$, $\lambda_4$, and $\lambda_5$. Our study involves individually adjusting the value of each parameter while keeping the remaining four fixed. Figure 4 presents the experimental results in terms of Mean Reciprocal Rank (MRR), where the x-axis represents $\log(\lambda_i)$, $i \in [1, 2, 3, 4, 5]$, and the y-axis represents MRR. Notably, a substantial value for $\lambda_1$ (*e.g.*, $\lambda_1 = 50$) tends to yield superior performance, highlighting the crucial role of the VAR model in capturing temporal dependencies among diverse system entities. For $\lambda_2$, MULAN achieves the best performance when $\lambda_2$ is set to 1, with a noticeable decline in performance as $\lambda_2$ increases. This suggests a delicate balance in the contribution of $\lambda_2$ to the overall model effectiveness. Regarding $\alpha_3$ and $\alpha_4$, optimal performance is observed when both are set to 20, underscoring their critical role in achieving superior results. Similarly, the best performance for MULAN aligns with a smaller value for $\lambda_5$, *e.g.*, $\lambda_5 = 0.1$. Further decreasing $\lambda_5$ results in a decline in performance, highlighting the importance of sparse regularization in the loss function.

*4.2.4 Ablation Study.* In this subsection, we conduct an ablation study to thoroughly assess the effectiveness of each component within the overarching objective function (Eq. 12). We consider four distinct variants of MULAN: MULAN-V, which excludes the VAR models responsible for modeling temporal dependencies among diverse system entities; MULAN-O, which disregards the orthogonal constraint ($\mathcal{L}_{orth}$); MULAN-N, which omits the extraction of modality-invariant information by excluding the node loss ($\mathcal{L}_{node}$) from the objective function; and MULAN-E, which removes the edge loss ($\mathcal{L}_{edge}$). We assess the performance of these variants in terms of Mean Reciprocal Rank (MRR), with performance evaluated by MAP@K available in Appendix B.5. A comparative analysis of MULAN against its variants consistently reveals performance degradation when any component is removed from the proposed method. For example, removing the edge loss induces a performance drop

of 16.7% and 23.3% on the Product Review and Online Boutique datasets, respectively. Similarly, excluding the node loss results in an 18.7% performance reduction on the Product Review dataset. These findings underscore the pivotal role of each component in ensuring the overall effectiveness of the proposed method.

## 5 RELATED WORK

**Root Cause Analysis**. Root cause analysis (RCA) is a systematic process aimed at uncovering the fundamental reasons for system failures using observed symptoms [31]. Numerous domain-specific RCA approaches [5, 9, 21, 24, 31–33, 42] have been developed to enhance the resilience of applications. Notably, in the context of the microservice systems, Li *et al.* [21] constructed a dependency graph based on system architecture knowledge and proposed a regression-based hypothesis testing to identify the root cause. Another study [16] integrated a hierarchical learning method with the PC algorithm, facilitating rapid identification of interventional targets. Additionally, [40] introduced a hierarchical graph neural networks-based algorithm to capture both intra-level and inter-level causal relationships for identifying the root causes in interdependent networks. Different from the existing works that primarily focus on unimodal data (*e.g.*, system metrics data), MULAN is a multimodal learning approach that extracts both modality-invariant and modality-specific information to enhance RCA performance.

**Multi-modal Learning**. Multi-modal learning has been extensively studied across various domains, such as natural language processing [11, 20], information retrieval [10, 25], and computer vision [22, 30, 48]. For example, within the domain of natural language processing, [11] explored the interplay among text, visual, and acoustic modalities to enhance sentiment analysis. In computer vision, numerous methods [17, 25] have been proposed to align text and image representations in the latent space. Nevertheless, in the context of RCA, the exploration of multi-modal RCA remains in a nascent stage. Research presented in [13, 46] has attempted to extract information from multi-modal data for root cause analysis. Regrettably, these studies primarily focus on each modality as a standalone entity, overlooking the intricate interconnections between them. In contrast, this paper systematically examines the interplay between different modalities, specifically between time-series data and unstructured text data, and co-constructs a comprehensive causal graph for root cause localization.

## 6 CONCLUSION

In this paper, we investigated the challenging problem of multimodal root cause localization in microservice systems. We proposed MULAN, a unified framework for localizing root causes by co-learning a causal graph from multi-modal data. MULAN leverages a log-tailored language model to facilitate causal graph generation from system logs. To explore the relationships among different modalities, both modality-invariant and modality-specific representations were extracted through node-level contrastive regularization and edge-level regularization. Additionally, we introduced a KPI-aware attention mechanism to assess modality reliability and facilitate the co-learning of the final causal graph. We validated the effectiveness of MULAN through extensive experiments on three real-world datasets. A promising direction for future work is extending MULAN to handle streaming data in an online setting.

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

# A EXTRACTING LABEL INFORMATION FOR LOG REPRESENTATION LEARNING

The extracting label information process involves partitioning the collected system logs into fixed time windows. For each of these windows, we create a log sequence, capturing unique log templates occurring within that specific time frame. In typical large language models, individual words in a sentence are treated as tokens. However, this approach isn't suitable for log sequence representation learning for two main reasons.

Firstly, the presence of a significant number of infrequent special tokens makes it challenging to learn effective representations for these tokens, given the limited sample size. Secondly, tokenizing log templates into lists of word tokens requires setting a very large maximal sequence length to accommodate all sequences within that limit. However, this can be problematic since, in practice, time windows are often set to 10 to 30 minutes to gather more valuable and reliable information. In some cases, the number of unique log templates exceeds 50, and when each word in these log templates is tokenized, the sequence length of the log sequence surpasses the default maximal sequence length (*e.g.*, 512) used in traditional large language models. When the sequence length exceeds this limit, the exceeding part is truncated, leading to information loss.

Conversely, having an extensive maximal sequence length poses multiple challenges. Firstly, it demands a substantial GPU memory and prolongs the training time, making it less feasible for practical deployment. This becomes particularly problematic when implementing an online system, which typically operates under tight time constraints. Online systems need to produce results before the arrival of the next batch of new data. Therefore, a large maximal sequence length becomes a hindrance to the deployment of online systems. Moreover, obtaining precise label information for log event templates presents a substantial challenge. The labeling process can be costly, and it often necessitates expert knowledge. The absence of high-quality label information poses a significant impediment to the ability of large language models to effectively learn the desired high-quality representations.

To address these challenges, our approach begins by capturing all unique log event templates within the specified time windows. We then transform the log sequence into a sequence of event template tokens using a specialized tokenizer. In this unique approach, each event template is treated as an individual token, as illustrated in Figure 2. In addition, different from the traditional large language, we also consider the frequency of each unique log template as the more frequently a log event template appears, the more important message it carries. This assumption proves highly valuable in addressing certain failure scenarios, such as Distributed Denial of Service (DDoS) attacks. During a DDoS attack, certain log event templates may experience a sudden, significant increase in frequency, signaling unusual behavior. Thus, we include the frequency information alongside each log event template, providing extra context to detect unusual patterns in potential failure cases.

To address the challenge of lacking label information, we propose two viable solutions. The first method is a "golden signal" approach that leverages domain knowledge. For instance, consider a microservice system, where system failures can be classified into various types, including DDoS attacks, storage failures, high CPU utilization, high memory utilization, and more. Each type of system failure is associated with specific keywords or "golden signals." By identifying these keywords within log event templates, we can determine whether a particular template is abnormal. These keywords may include terms like "error," "exception," "critical," "fatal," "timeout," "connection refused," "No space left on the device," "out of memory," "terminated unexpectedly," "backtrace," "stack trace," "service unavailable," "502 Bad Gateway," "503 Service Unavailable," "504 Gateway Timeout," "unable to connect to," "rate limit exceeded," "request limit exceeded," "cloud system down," "cloud service not responding," "failure," "corrupted data," "data loss," "file not found," "high CPU utilization," "CPU spike," "CPU saturation," "excessive CPU usage," "failed," "shutdown," "Permission denied," "DEBUG," and more. The presence and extent of these abnormal log event templates within a log sequence are measured to compute the overall abnormality of the sequence, which serves as label information. When domain knowledge is not readily available, our second solution involves using anomaly detection models, such as Deeplog [8] or OC4Seq [43], to evaluate the abnormality of a log sequence.

# B ADDITIONAL EXPERIMENT

## B.1 Implementation Details

All experiments are conducted on a server running Ubuntu 18.04.5 with an Intel(R) Xeon(R) Silver 4110 CPU @2.10GHz and a 4-way 11GB GTX2080 GPU.

## B.2 Evaluation Metric

We evaluate the model performance with the following three widely-used metrics [24, 39]:

(1). **Precision@K (PR@K)**: It measures the probability that the top $K$ predicted root causes are real, defined as:

$$PR@K = \frac{1}{|\mathbb{A}|} \sum_{a \in \mathbb{A}} \frac{\sum_{i<k} R_a(i) \in V_a}{\min(K, |v_a|)} \quad (15)$$

where $\mathbb{A}$ is the set of system faults, $a$ is one fault in $\mathbb{A}$, $V_a$ is the real root causes of $a$, $R_a$ is the predicted root causes of $a$, and i is the $i$-th predicted cause of $R_a$.

(2). **Mean Average Precision@K (MAP@K)**: It assesses the top $K$ predicted causes from the overall perspective, defined as:

$$MAP@K = \frac{1}{K|\mathbb{A}|} \sum_{a \in \mathbb{A}} \sum_{i \le j \le K} PR@j \quad (16)$$

where a higher value indicates a better performance.

(3). **Mean Reciprocal Rank (MRR)**: It evaluates the ranking capability of models, defined as:

$$PR@K = \frac{1}{|\mathbb{A}|} \sum_{a \in \mathbb{A}} \frac{1}{rank_{R_a}} \quad (17)$$

where $rank_{R_a}$ is the rank number of the first correctly predicted root cause for system fault $a$.

## B.3 How to Choose High-quality and Low-quality System Metrics?

In the experiment, we first measure the performance of each single-modality baseline method by only using one single system metric (*e.g.*, CPU usage, memory usage, rate transmit rate, *etc*). Then, we

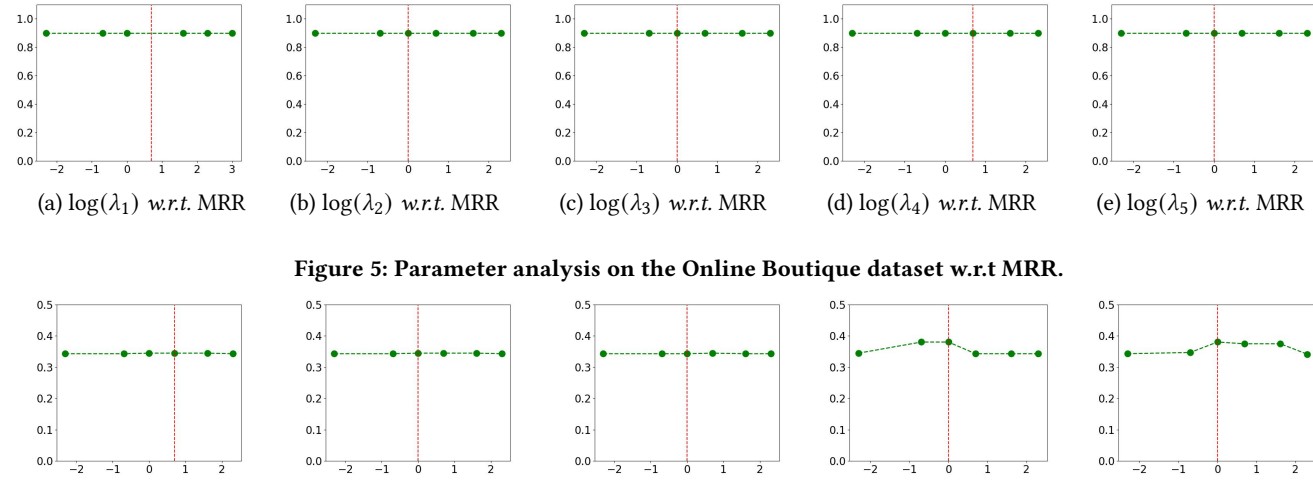

(a) $\log(\lambda_1)$ *w.r.t.* MRR   (b) $\log(\lambda_2)$ *w.r.t.* MRR   (c) $\log(\lambda_3)$ *w.r.t.* MRR   (d) $\log(\lambda_4)$ *w.r.t.* MRR   (e) $\log(\lambda_5)$ *w.r.t.* MRR

**Figure 5: Parameter analysis on the Online Boutique dataset w.r.t MRR.**

(a) $\log(\lambda_1)$ *w.r.t.* MRR   (b) $\log(\lambda_2)$ *w.r.t.* MRR   (c) $\log(\lambda_3)$ *w.r.t.* MRR   (d) $\log(\lambda_4)$ *w.r.t.* MRR   (e) $\log(\lambda_5)$ *w.r.t.* MRR

**Figure 6: Parameter analysis on Train Ticket dataset w.r.t MRR.**

**Table 7: Ablation study on three datasets evaluated by MAP@K.**

| Model | Product Review | Online Boutique | Train Ticket |
|-------|----------------|-----------------|--------------|
| - | MAP@10 | MAP@5 | MAP@10 |
| MULAN | **1.0** | **0.96** | **0.386** |
| MULAN-V | 0.98 | 0.92 | 0.385 |
| MULAN-O | 0.96 | **0.96** | 0.357 |
| MULAN-N | 0.94 | 0.88 | 0.371 |
| MULAN-E | 0.96 | 0.84 | 0.357 |

**Table 6: Quality measurement of high-quality metric and low-quality metric on Product Review dataset. The median ranking scores are used to evaluate the quality of different metrics. The best metric is denoted as High-quality while the worst metric is denoted as Low-quality.**

| Metric | Case 1 | Case 2 | Case 3 | Case 4 |
|--------|--------|--------|--------|--------|
| High-quality | 21 | 30 | 11 | 25 |
| Low-quality | 82 | 68 | 30 | 40 |

select the system metric with the highest median ranking score as the high-quality system metric denoted $M^+$, and the system metric with the lowest median ranking score as the low-quality system metric denoted $M^-$. The ranking results are shown in Table 6.

### B.4 Additional Parameter Analysis

In this subsection, we delve into an analysis of parameter sensitivity within the MULAN framework on the Online Boutique and Train Ticket datasets, specifically examining the impact of variations in $\lambda_1, \lambda_2, \lambda_3, \lambda_4$, and $\lambda_5$. Figure 5 and Figure 6 present the experimental results with respect to Mean Reciprocal Rank (MRR), where the x-axis is $\log(\lambda_i), i \in [1, 2, 3, 4, 5]$ and the y-axis is MRR. By observations on Figure 5, we find that the value of $\lambda_1, \lambda_2, \lambda_3, \lambda_4$, and $\lambda_5$ does not influence the performance of MULAN on Online Boutique dataset. Our conjecture for this observation is that the number of system entities is only 10 and it's an easy task to identify the root cause by our method. Based on the experimental results on the Train Ticket dataset, we found that the change of the values for $\lambda_1, \lambda_2$, and $\lambda_3$ do not have a great impact on the performance of MULAN. MULAN achieves the best result with $\lambda_5 = 1$ on the Train Ticket dataset.

### B.5 Additional Results for Case Study

Table 7 shows the performance evaluated by MAP@K. Comparing the performance of MULAN with other variants, removing any component of the proposed methods consistently results in performance degradation. For instance, removing the edge loss causes the performance to drop by 12% on the Online Boutique dataset while removing the node loss leads to a 6% performance reduction on the Product Review dataset.

