# OpenReview forum: "MULAN: Multi-modal Causal Structure Learning and Root Cause Analysis for Microservice Systems"
_ACM.org/TheWebConf/2024/Conference — TheWebConf24 Oral_

### Official Review · Reviewer_rsu1 · 2023-11-24

**Novelty:** 5
**Technical Quality:** 5

**Review:**

This work addresses the causal structure learning problem in multi-modal settings. To solve this problem, the paper proposes a novel method MULAN, which can identify root causes in microservice systems. In this method, a language model is used in this method to convert log into time-series data; a contrastive learning approach is used to learn modality-invariant and modality-specific representations; an attention mechanism is used to assess modality reliability and learn a causal graph.

Pros:
+ This studied problem is important and has potential for real-world application.
+ The proposed method is novel in leveraging multiple modalities.
+ The experiments on three real-world datasets show the effectiveness of the method.

Cons:
+ It is not quite clear how to measure the overall performance of causal discovery.
+ Need further justification w.r.t. details in the experiments.
+ Could be better if complexity could be provided.

Generally, the paper proposes a novel method with reasonable technical design and experiments. But it would be better if the above concerns could be addressed.

**Questions:**

1. The experimental metrics seem to only cover root cause identification and predicted causes. Is there any metric to evaluate the correctness of the whole causal graph learned by the proposed method?

2. Can the proposed method still be applied in single-modality cases? If so, can the authors provide a comparison between the proposed method and other baselines in the experiments?

3. How scalable is the proposed method in large-scale data? What is the complexity?

**Reviewer Confidence:**

3: The reviewer is confident but not certain that the evaluation is correct

**Scope:**

4: The work is relevant to the Web and to the track, and is of broad interest to the community

---

### Official Review · Reviewer_wmj2 · 2023-11-27

**Novelty:** 4
**Technical Quality:** 5

**Review:**

The paper introduces MULAN, which leverages multi-modal data, employing a log-tailored language model for log representation, a contrastive learning approach for extracting modality-invariant and specific representations, a key performance indicator-aware attention mechanism, and network propagation for root cause localization. The method's effectiveness is demonstrated through experiments on three real-world datasets

Strength:
	1. MULAN's integration of multiple data modalities is a significant advancement. By not limiting itself to a single data source, it captures more intricate system relationships, enhancing RCA's accuracy and reliability.
	2. The development of specific modules like the log-tailored language model and KPI-aware attention mechanism represents a novel approach to RCA. These modules help in accurately capturing the intricacies of microservice systems, which is crucial for effective RCA.
	3. The application and validation of MULAN on three diverse real-world datasets demonstrate its practical applicability and robustness

Weakness:
1. The paper lacks a clear delineation of its contributions in the context of previous works that also use multi-modality input.
2. The paper needs a clearer explanation of the motivation behind separating modality-specific and modality-invariant representations and then fusing them.

I recommend accepting the paper, Its innovative approach to multi-modal data analysis for root cause analysis significantly contributes to the field. These contributions outweigh the weaknesses, which can be addressed in future work.

**Questions:**

1. What motivated the decision to modality-specific and modality-invariant representations process pipeline in MULAN, and how does this approach enhance root cause analysis effectiveness?
2. Can you elaborate on the importance and impact of modality-specific features relative to modality-invariant features in MULAN's analysis? How does parameter α from Equation 4 influence this relationship? Is there any real-world concept that corresponds to modality-specific representations or modality-invariant representations? The explanation of this can help the reader to understand the effort paid to separate them.
3. How does MULAN's methodology distinctly contribute to the field of RCA in comparison to similar multi-modality-based works like“Eadro: An End-to-End Troubleshooting Framework for Microservices on Multi-source Data”, “Robust Failure Diagnosis of Microservice System through Multimodal Data”.

**Reviewer Confidence:**

3: The reviewer is confident but not certain that the evaluation is correct

**Scope:**

4: The work is relevant to the Web and to the track, and is of broad interest to the community

---

### Official Review · Reviewer_Z9rj · 2023-12-05

**Novelty:** 6
**Technical Quality:** 6

**Review:**

This paper investigated the problem of multimodal root cause localization in microservice systems. It proposed MULAN, a unified framework for localizing root causes by co-learning a causal graph from multi-modal data. This work includes four key modules: (1) representation extraction via logtailored language model; (2) contrastive multi-modal causal structure learning; (3) causal graph fusion with KPI-aware attention; and (4) network propagation based root cause localization.

I believe this topic is popular and people in industries are waiting for good solutions to the RCA studies. The idea of chaining multi-modals with causal structure learning should be a good solution. And the approach is reasonable to have these four steps/modules chained, especially the graph fusion part. I like this work.

And the experiments are complete and I can find results that would raise questions, such as single-modality scenarios and the ablation study. These experiments do help a lot to persuade readers that each module contributes to the final performance.

The only issue I have is that there are few discussions of insights on where the key changes/differences that make this outperform others. Is it really because of this work "interplay between different modalities"? And others not?

**Questions:**

1. I still cannot find the contribution of "Representation Extraction via Log-tailored Language Model". How does this part contribute to the final performance? And I haven't found experiments about this. I see it said, "To enable multi-modality modeling, we initially convert the system logs into time-series data using the Regression-based language model introduced in Section 3.1." So, is it also included in other methods in the experiment?

2.  Is the graph fusion part (module 3) the key to outperforming other methods?

3. What is the most time-consuming module? I guess except for the first preprocessing, other modules should be done very quickly.

**Reviewer Confidence:**

2: The reviewer is willing to defend the evaluation, but it is likely that the reviewer did not understand parts of the paper

**Scope:**

4: The work is relevant to the Web and to the track, and is of broad interest to the community

---

### Decision · Program_Chairs · 2024-01-22

**Decision:**

Accept (Oral)

**Comment:**

The paper presents a multi-modal causal data analysis approach called MULAN for studying root cause localization method in microservice networks. The approach novelly integrates several techniques, including a log-tailored language model for log representation, a contrastive learning approach for extracting modality-invariant and specific representations, a key performance indicator-aware attention mechanism, and random walk for network propagation. The method is empirically showing promising results.